# Comparative Impact of Integrated Palliative Care vs. Standard Care on the Quality of Life in Cancer Patients: A Global Systematic Review and Meta-Analysis of Randomized Controlled Trials

**Addisu Getie**[1]*, **Afework Edmealem**[1], **Tegene Atamenta Kitaw**[2]

**1** Department of Nursing, College of Medicine and Health Sciences, Debre Markos University, Debre Markos, Ethiopia, **2** Department of Nursing, College of Medicine and Health Sciences, Woldia University, Woldia, Ethiopia

* addisu_getie@dmu.edu.et

## Abstract

### Introduction

Cancer is a leading cause of global morbidity and mortality, significantly impairing patients' quality of life (QoL). Integrated Palliative Care (IPC) has been proposed as a holistic approach to enhance quality of life by addressing patients' physical, emotional, and psychosocial needs. While some studies suggest Integrated Palliative Care improves quality of life more than standard care, the evidence remains inconclusive. This systematic review and meta-analysis aim to evaluate the comparative impact of Integrated Palliative Care versus standard care on the quality of life in cancer patients.

### Methods

A comprehensive search of databases including PubMed, Cochrane Library, and Embase was conducted. We selected randomized controlled trials (RCTs) comparing Integrated Palliative Care and standard care for cancer patients, focusing on the quality of life as measured by validated tools such as the EORTC QLQ-C30 and FACT-G. Data were pooled using a random-effects model to account for study heterogeneity. Subgroup and sensitivity analyses were also performed.

### Results

Nine randomized controlled trials involving 1,794 patients met the inclusion criteria. Meta-analysis showed that Integrated Palliative Care significantly improved quality of life compared to standard care (SMD = 3.25; 95% CI: 1.20–5.30; p < 0.001). Studies conducted in Asia showed the highest standardized mean difference (SMD = 6.15; 95% CI: 3.07–9.23; p < 0.001), followed closely by studies from Africa (SMD = 6.0; 95% CI: 5.13–6.87; p < 0.001), compared to those from other regions. Similarly, research focusing on lung cancer patients showed the greatest standardized mean difference of (SMD = 6.15; 95% CI: 3.07–9.23; p < 0.001) relative to other cancer types. Furthermore, studies

**Data availability statement:** All relevant data are within the article and its Supporting Information files.

**Funding:** The author(s) received no specific funding for this work.;

**Competing interests:** The authors have declared that no competing interests exist

**Abbreviations CI:** Confidence IntervalICP:Integrated Palliative CareQoL:Quality of LifePRISMA:Preferred Reporting Items for Systematic Review and Meta-analysisRCTs:Randomized Control TrialsSMD:Standardized Mean Difference

involving newly diagnosed cancer patients recorded the highest standardized mean difference of (SMD = 5.69; 95% CI: 4.57–6.80; p < 0.001).

## Conclusion

Integrated Palliative Care significantly enhances the quality of life in cancer patients compared to standard care. These findings support integrating Integrated Palliative Care into oncology practices to provide comprehensive, patient-centered care that addresses both physical and emotional needs. Further research should explore long-term benefits across diverse populations.

## Introduction

Cancer remains one of the leading causes of morbidity and mortality worldwide, profoundly impacting patients' quality of life (QoL) [1]. With an increasing incidence of cancer cases, the need for effective care strategies that address not only the medical but also the psychosocial and emotional needs of patients has become paramount. Integrated Palliative Care (IPC) is a multi-disciplinary approach to palliative care that combines medical, psychological, social, and spiritual support to enhance the quality of life for cancer patients. This model involves collaboration among healthcare providers, including oncologists, palliative care specialists, nurses, social workers, and spiritual care providers, to deliver comprehensive care that addresses the physical, emotional, and social needs of patients. Standard care refers to the usual treatment provided for cancer patients, which typically involves conventional cancer therapies such as chemotherapy, radiotherapy, and surgery, without the integration of specialized palliative care services [2,3]. Integrated palliative care (IPC) has emerged as a comprehensive approach designed to improve the QoL of cancer patients by providing multidisciplinary support and addressing various aspects of their well-being. By integrating symptom management, psychosocial support, and advanced care planning, IPC aims to enhance the overall experience of patients during the continuum of care [4].

Research indicates that IPC can significantly alleviate pain, anxiety, and depression, which are prevalent among cancer patients [5]. However, the effectiveness of IPC compared to standard care in enhancing QoL remains a topic of debate. Standard care typically focused on curative treatment, often overlooks the holistic needs of patients, which may lead to suboptimal patient experiences and increased distress [6]. Thus, there is a critical need to evaluate the comparative impact of IPC versus standard care on QoL outcomes among cancer patients through systematic analysis of randomized controlled trials (RCTs).

Despite the growing body of literature on the subject, previous studies have yielded mixed results, highlighting a gap in consensus regarding the efficacy of IPC in improving QoL [7]. Several studies have demonstrated that integrated palliative care (IPC) significantly improves patient outcomes, particularly in areas such as symptom management and emotional support [8]. A systematic review and meta-analysis of RCTs will help to consolidate the available evidence and provide clearer insights into the potential benefits of IPC over standard care in this patient population.

This systematic review builds on the growing body of literature that has explored the impact of Integrated Palliative Care (IPC) on the quality of life (QoL) of cancer patients, acknowledging the important contributions of earlier studies. While prior research has provided valuable insights, many have focused on specific populations or small sample sizes, and their findings have been mixed, leaving some gaps in understanding. Our study seeks to address these gaps by synthesizing evidence from the most recent randomized controlled trials (RCTs), which offer more rigorous and reliable data. By including a broader range of RCTs, this review aims to provide an updated, comprehensive analysis of the comparative impact of IPC versus standard care across diverse cancer types and settings.

The unique contribution of our study lies in its ability to incorporate the latest evidence, allowing for a more nuanced understanding of the outcomes of IPC on QoL in cancer patients. We also explored variations in the effectiveness of IPC based on different parameters including regional contexts, cancer site, and patient category providing a global perspective on its role in palliative care. By synthesizing this contemporary data, this review offered robust and evidence-based conclusions that can directly inform clinical practice, healthcare policies, and the integration of IPC into routine oncology care. Ultimately, we hope to further establish IPC as an essential component in improving the QoL of cancer patients and ensure that its integration into standard oncology practice is strongly supported by the latest research.

## Method

### The study protocols

The results of this systematic review and meta-analysis were reported using the Preferred Reporting Items for Systematic Review and Meta-analysis (PRISMA) reporting guideline (S1 Table in S1 Text).

### Search strategy

For this systematic review and meta-analysis, a comprehensive search strategy was employed to identify relevant randomized controlled trials (RCTs) comparing the impact of integrated palliative care with standard care on the quality of life in cancer patients. The search was conducted across multiple databases including PubMed, MEDLINE, Embase, CINAHL, Cochrane Library, and Web of Science. Search terms included a combination of keywords and Medical Subject Headings (MeSH) related to palliative care, standard care, cancer, quality of life, and randomized controlled trials. Boolean operators (AND, OR) were used to refine the search, and filters were applied to limit the studies to RCTs published in English for the past ten years. Additionally, references of selected articles were hand-searched to identify any further relevant studies. Duplicates were removed, and the selection process was following the PRISMA guidelines. Articles published until September 2024 were included in the search. The last date for searching was September 15, 2024

### Eligibility criteria

For this systematic review and meta-analysis, the inclusion criteria were centered on randomized controlled trials (RCTs) that had compared the effects of integrated palliative care with standard care in cancer patients, with a particular emphasis on evaluating their impact on patients' quality of life. We included only RCTs because they are regarded as the gold standard for assessing the effectiveness of interventions, offering high-quality evidence with a low risk of bias. By focusing on RCTs, we sought to ensure the reliability and strength of our findings, thereby strengthening the validity of our conclusions about the impact of integrated palliative care on the quality of life in cancer patients. Studies were included if they involved adult cancer patients of any cancer type and provided quantitative data on quality of life using standardized, validated assessment tools. Only peer-reviewed studies published in English within the last ten years, from any geographical region, were considered. The exclusion criteria, on the other hand, ruled out non-randomized studies, including observational studies, case reports, conference proceedings, and review articles. Additionally, studies that did not specifically compare integrated palliative care to standard care or failed to report measurable quality-of-life outcomes were excluded. Furthermore, trials that exclusively

focused on pediatric cancer patients, those involving non-cancer populations, or those that were not available in full-text format were also excluded from the review.

## Outcome measurement of the study

The outcome measurement of this global systematic review and meta-analysis of randomized controlled trials focused on the quality of life (QoL) in cancer patients, as assessed by standardized and validated instruments, such as the European Organization for Research and Treatment of Cancer Quality of Life Questionnaire (EORTC QLQ-C30) and the Functional Assessment of Cancer Therapy-General (FACT-G) scale. These tools were consistently used across studies to capture comprehensive dimensions of QoL, including physical, emotional, and social well-being. Secondary outcomes, including symptom management (e.g., pain, fatigue, and psychological distress), were evaluated using tools like the Edmonton Symptom Assessment System (ESAS) to provide a detailed understanding of patient-reported symptom burden.

## Data collection and quality assessment

For the systematic review and meta-analysis titled "Comparative Impact of Integrated Palliative Care vs. Standard Care on the Quality of Life in Cancer Patients: A Global Systematic Review and Meta-Analysis of Randomized Controlled Trials," data collection was conducted by systematically searching multiple electronic databases, including PubMed, Cochrane Central Register of Controlled Trials (CENTRAL), and Embase, for randomized controlled trials (RCTs) published in the last ten years. The search was supplemented by manual screening of reference lists from relevant studies to ensure comprehensive coverage. Studies that met predefined inclusion criteria, such as those involving cancer patients receiving either integrated palliative care or standard care, were selected. Data were extracted using a standardized form, capturing study characteristics, interventions, outcomes, and quality of life measures. The Cochrane Risk of Bias Tool was employed to assess the methodological quality of included trials, focusing on randomization processes, blinding, and completeness of outcome data. Studies were categorized as high, moderate, or low risk of bias based on these assessments. A study is categorized as high risk if there are significant concerns in any of the key areas (randomization, blinding, completeness of data) that could potentially impact the study's results. It is classified as a moderate risk if there are some issues, but these are unlikely to fully compromise the findings. A study is considered low risk if it is methodologically robust and free from major biases. The quality of the included randomized controlled trials (RCTs) was evaluated using the Joanna Briggs Institute (JBI) Critical Appraisal Checklist, focusing on key aspects of study design such as randomization, allocation concealment, blinding, and follow-up completeness. This ensured a systematic and rigorous assessment of methodological quality [9]. Data extraction and quality assessment were performed independently by two reviewers to minimize bias, with disagreements resolved through consensus or consultation with a third reviewer. Additionally, we used Covidence software to facilitate the screening and selection of studies in our systematic review. The use of this tool helped streamline the process and ensured a more efficient and transparent review. This rigorous process ensured the reliability and validity of the findings, adhering to recent guidelines for conducting systematic reviews and meta-analyses in health care research [10,11].

## Data synthesis and analysis

The overall global comparative impact of integrated palliative care vs. standard care on the quality of life in cancer patients was measured using the standardized mean difference and

pooled using a weighted inverse variance random-effects model at 95%CI [12]. The data were extracted and cleaned using Microsoft Excel spreadsheets and exported to STATA version 11.0 (Stata Corporation, College Station, Texas) software for analysis. The heterogeneity of the studies was assessed using the Cochrane Q test and $I^2$ with its corresponding p-value [13]. To examine the source of heterogeneity, subgroup analysis, sensitivity analysis, and meta-regression were carried out. In addition, the presence of publication bias was evaluated by using Egger's test and funnel plot [14]. Finally, a statistical test with a P-value of less than 0.05 was considered statistically significant.

### Ethics approval and consent to participate

Not applicable.

## Results

In this systematic review and meta-analysis, a comprehensive search was conducted across multiple databases using targeted keywords, yielding a total of 1,141 RCTs. However, a large portion of these studies was excluded during the screening process for various reasons. Many were eliminated due to duplication, failure to report outcomes relevant to the study, inadequate methodological quality, or lack of access to the full text. Furthermore, after a detailed evaluation of titles and abstracts, additional RCTs were excluded as they did not meet the inclusion criteria. Following this rigorous screening process, a final selection of nine RCT studies was included for final analysis, as they satisfied all the criteria for inclusion in the review (Fig 1).

### Study characteristics

Nine randomized controlled trials involving 1,794 patients met the inclusion criteria conducted globally, with studies spanning Europe [15–20], Africa [21], North America [22], and Asia [23] between 2015 and 2023 were included. These trials evaluated the comparative impact of integrated palliative care (IPC) versus standard care (SC) on the quality of life (QoL) in cancer patients. The types of cancers varied widely, including lung, gastrointestinal, breast, and reproductive organ cancers, with patient populations encompassing both newly diagnosed and patients on follow-up. Sample sizes across the studies ranged from 57 to 280 participants, providing a robust comparison between IPC and SC groups. Quality of life outcomes were commonly assessed using validated tools, and the follow-up periods ranged from several months to over a year. The studies consistently showed a positive impact of IPC on patient satisfaction, symptom management, and overall quality of life, especially in patients who were on follow up, supporting the integration of palliative care into standard treatment protocols globally (Table 1).

### Risk of bias and quality assessment

We conducted a comprehensive risk of bias and quality assessment for the included randomized controlled trials (RCTs) using the Joanna Briggs Institute (JBI) Critical Appraisal Checklist and the Cochrane Risk of Bias (RoB 2) tool, represented with a traffic light system. The JBI checklist was employed to evaluate study design aspects, including randomization, allocation concealment, blinding, and completeness of follow-up, ensuring a structured appraisal of methodological rigor (S2 Table in S1 Text). The Cochrane RoB 2 tool provided a detailed assessment across domains such as bias in randomization, deviations from intended interventions, missing outcome data, measurement of outcomes, and selection of reported results.

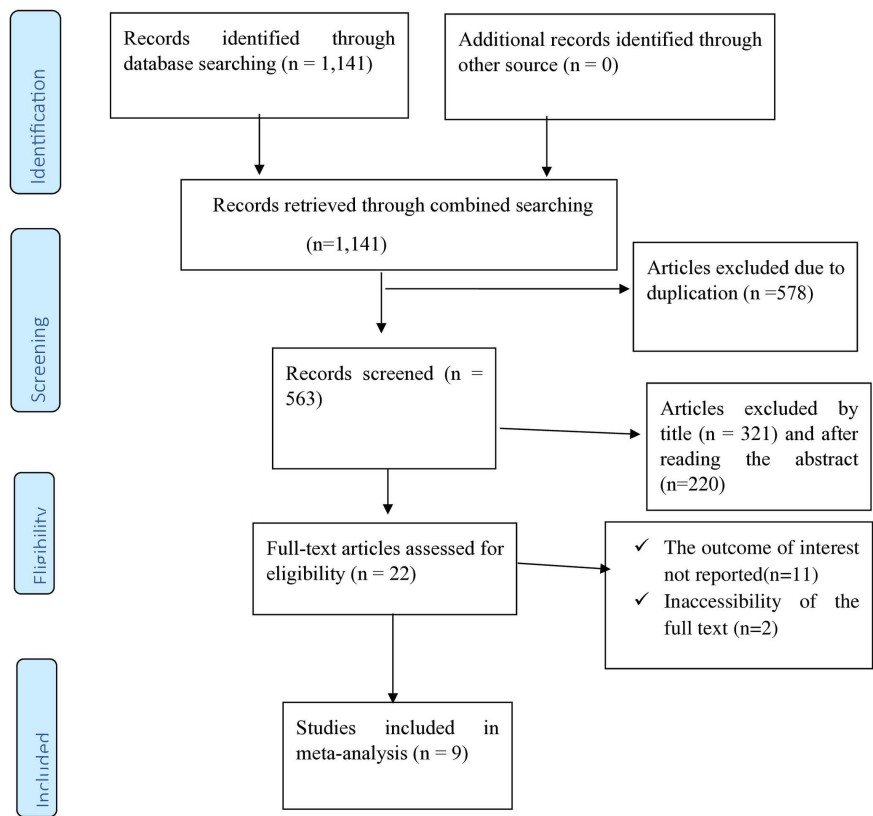

**Fig 1. PRISMA flow chart diagram on a selection of RCTs done on the comparative impact of integrated palliative care vs. standard care on the quality of life in cancer patients.**

Each domain was categorized as "low risk," "some concerns," or "high risk," and visually summarized using traffic light plots to highlight the distribution of risk levels (S1 Fig in S1 Text). This dual approach allowed for a robust evaluation, ensuring transparency and reliability in interpreting the quality of the evidence base.

## Meta-analysis

**Impact of integrated palliative care.** The study comparing the impact of integrated palliative care versus standard care on the quality of life in cancer patients showed a significant improvement in the integrated palliative care group, with a standard mean difference (SMD = 3.25; 95% CI: 1.20–5.30; p < 0.001). This indicates that patients receiving integrated palliative care experienced a notable enhancement in their quality of life compared to those receiving standard care (Fig 2).

**Heterogeneity and investigation of the source of heterogeneity.** Due to the presence of heterogeneity ($I^2$ =82.94%) among the included studies, a subgroup analysis, meta-regression, and sensitivity analysis were conducted to explore the potential source of variations. The moderators considered to conduct subgroup analysis and meta-regression were the continent where the studies were conducted, the site of cancer, the patient category, and follow up duration. These analyses aimed to identify whether the observed variations in the outcomes could be attributed to these factors, providing a clearer understanding of the differential impact of the interventions across diverse populations and cancer types. A subgroup analysis was conducted based on the study settings, categorizing the studies by their respective regions:

**Table 1. Study characteristics on RCTs done to evaluate the comparative impact of integrated palliative care vs. standard care on the quality of life in cancer patients.**

| Author and publication year | Study setting | Site of cancer | Patient category | Follow up duration in month | Sample size of IG | Sample size of CG | Mean QoL of cancer patients who received integrated palliative care | Mean QoL of cancer patients who received standard care | SD QoL of cancer patient who received integrated palliative care | SD QoL of cancer patient who received standard care | Name of data extractor | Date of data extraction | Eligibility to be included based on JBI |
|---|---|---|---|---|---|---|---|---|---|---|---|---|---|
| Franciosi V et al, 2019 | Europe | Lung and GI | On follow up | 3 | 113 | 105 | 70.1 | 69.6 | 15.5 | 15.5 | Addisu Getie | Sep.3, 2024 | Yes |
| Marie A et al, 2015 | Europe | Breast, GI, reproductive organ | On follow up | 3 | 29 | 28 | 12.6 | 11.7 | 1.9 | 1.9 | Addisu Getie | Sep. 5, 2024 | Yes |
| Reid EA et al, 2022 | Africa | All | Newly diagnosed | 3 | 42 | 53 | 24 | 18.0 | 2.5 | 1.8 | Addisu Getie | Sep.6, 2024 | Yes |
| Vanbutsele G et al, 2018 | Europe | All | On follow up | 3 | 92 | 94 | 61.98 | 54.4 | 24 | 25.3 | Addisu Getie | Sep.8, 024 | Yes |
| Chen M et al, 2023 | Asia | Lung only | Newly diagnosed | 6 | 140 | 140 | 117.81 | 111.7 | 11.15 | 14.9 | Addisu Getie | Sep.11, 2024 | Yes |
| Groenveld M et al, 2017 | Europe | All | On follow up | 2 | 145 | 152 | 57.6 | 59.9 | 22.3 | 22.5 | Afework Edmealem | Sep.11, 2024 | Yes |
| Slama O et al, 2020 | Europe | All | On follow up | 6 | 60 | 66 | 66.7 | 62.8 | 25.4 | 25 | Afework Edmealem | Sep.13, 2024 | Yes |
| Jennifer S et al, 2016 | North America | Lung and GI | Newly diagnosed | 3 | 175 | 175 | 81 | 77.7 | 13.3 | 12.9 | Afework Edmealem | Sep.14,2024 | Yes |
| Vanbutsele G et al, 2020 | Europe | Breast, GI, reproductive organ | On follow up | 3 | 91 | 94 | 61 | 58.0 | 18.6 | 20.1 | Afework Edmealem | Sep.15,2024 | Yes |

CG: Control Group, IG: Intervention Group, GI: Gastrointestinal, JBI: Joanna Briggs Institute, SD: Standard Deviation, all: RCTs that included cancer patients of various types, with the study populations consisting of cancer patients regardless of the cancer site.

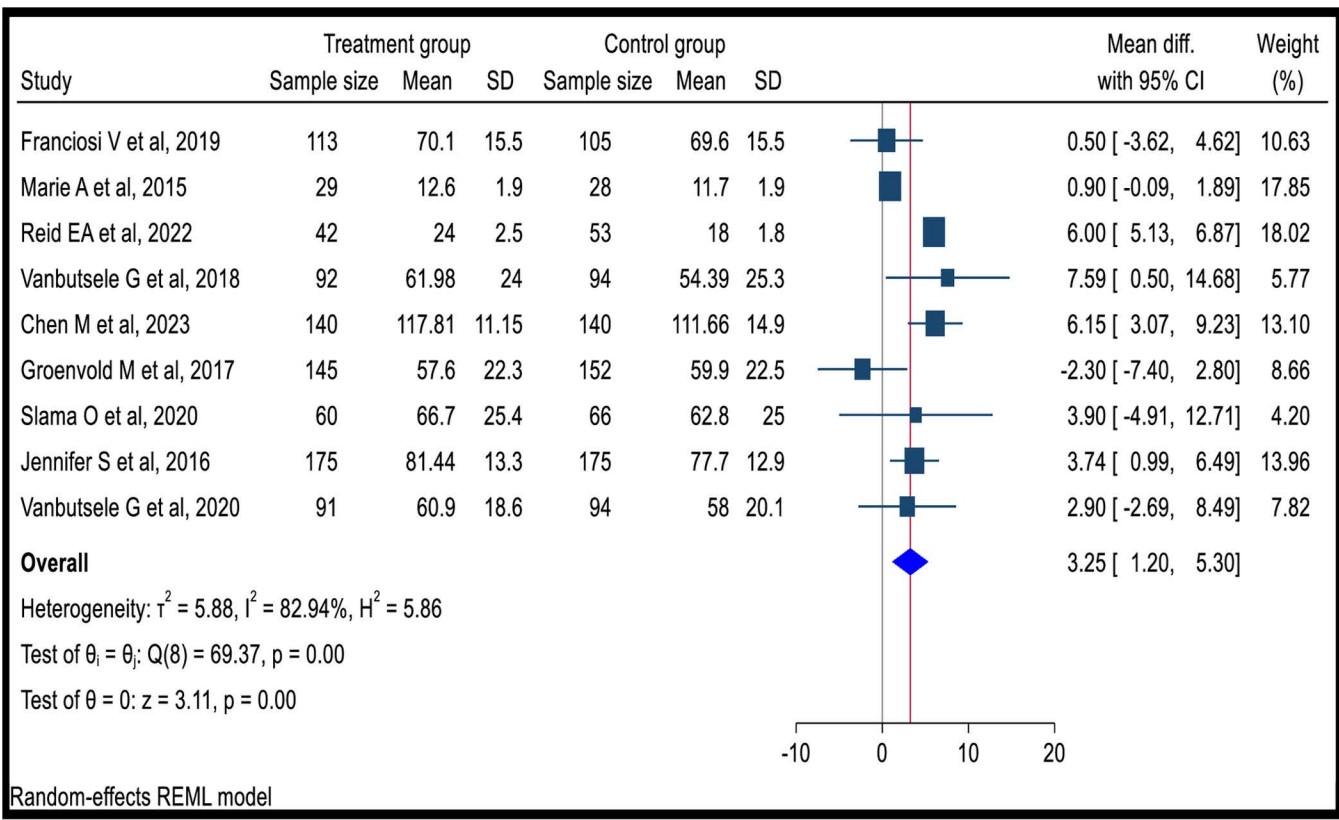

**Fig 2. Forest plot using a random-effects model illustrating the comparative effects of integrated palliative care versus standard care on the quality of life in cancer patients.**

Europe, Africa, Asia, and North America. Thus, studies from Asia reported the highest standardized mean difference (SMD = 6.15; 95% CI: 3.07–9.23; p < 0.001), followed closely by studies conducted in Africa (SMD = 6.00; 95% CI: 5.13–6.87; p < 0.001), compared to those from other continents (Fig 3). Similarly, a subgroup analysis was conducted based on cancer type. The analysis revealed that studies involving lung cancer patients exhibited the highest standardized mean difference (SMD = 6.15; 95% CI: 3.07–9.23; p < 0.001) (Fig 4). Additionally, a subgroup analysis was performed based on patient categories, specifically newly diagnosed cancer patients and those on follow-up. Studies involving newly diagnosed cancer patients showed the highest standardized mean difference (SMD = 5.69; 95% CI: 4.57–6.80; p < 0.001) (Fig 5). Furthermore, subgroup analysis was conducted based on the duration of follow-up, which included follow-up periods of two, three, and six months. Studies with six months of follow-up period showed the highest standardized mean difference (SMD = 5.90; 95% CI: 2.99–8.81) (Fig 6). The findings indicate that integrating palliative care has a significantly greater impact on patients from Asia and Africa, as well as those with lung cancer and individuals who are newly diagnosed. The meta-regression results indicated that the moderators, continent, site of cancer, and follow up duration, were not significant sources of heterogeneity, with P-values of 0.261, 0.58, 0.231, respectively. However, the patient category emerged as a potential source of heterogeneity, as indicated by a P-value of 0.019. The sensitivity analysis, conducted using a one-point leave-out method, revealed no influential studies, as all remained within the confidence interval of 1.20–5.30. This suggests that the overall results are robust, and no single study disproportionately influenced the findings (Fig 7).

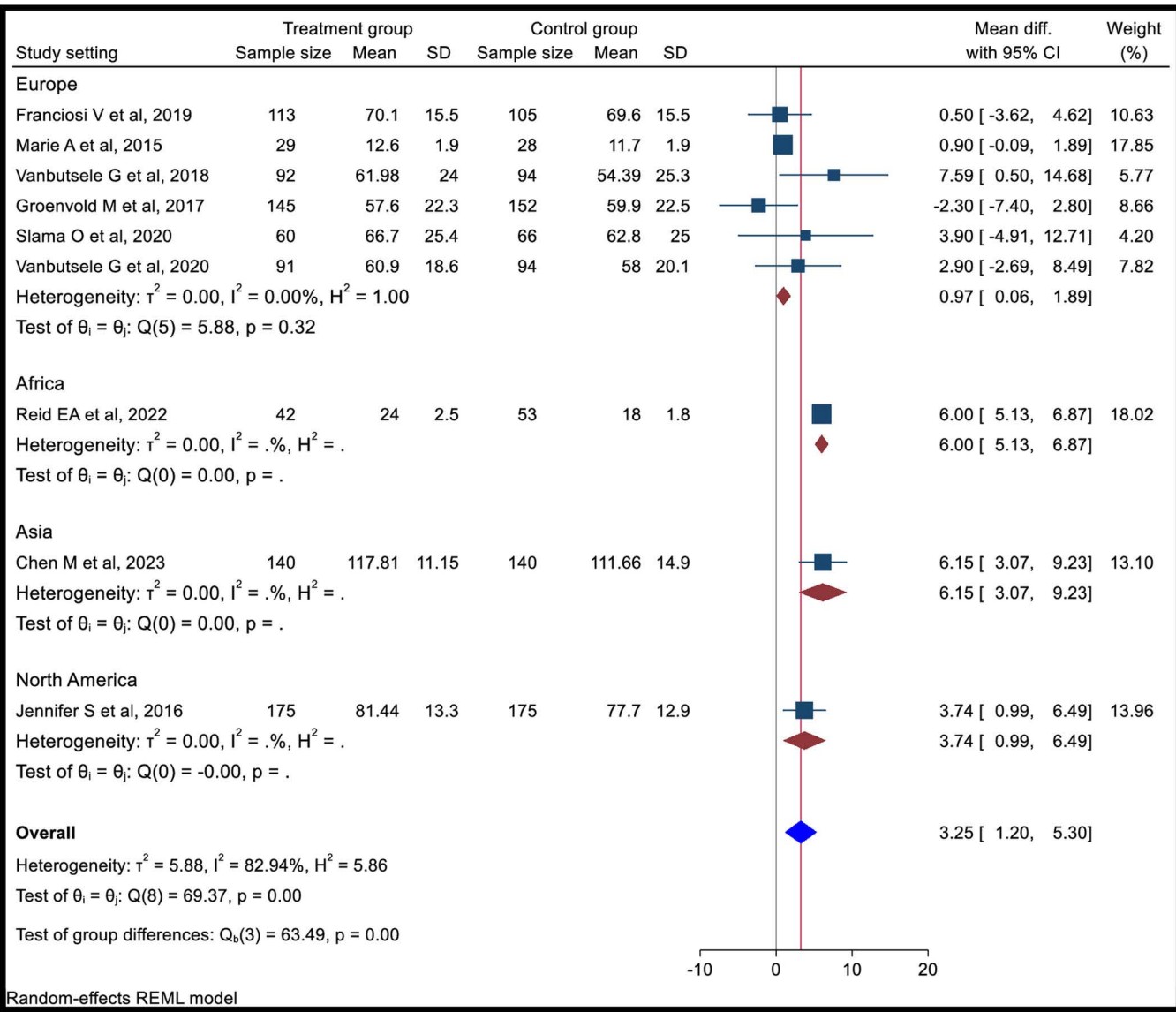

**Fig 3. Forest plot using a random-effects model illustrating sub-group analysis by continent where RCTs were conducted to show the comparative effects of integrated palliative care versus standard care on the quality of life in cancer patients.**

**Publication bias.** The funnel plot demonstrated a symmetrical distribution of the included studies, indicating the absence of significant publication bias. Additionally, Egger's test yielded a non-significant result (P = 0.8775), further confirming that there is no strong evidence of bias affecting the overall findings (Fig 8). The trim and fill analysis confirmed the absence of publication bias, as the mean difference between the observed studies and the observed plus imputed studies remained consistent. There was no significant change in the mean difference, further validating the robustness of the results and the lack of bias in the included studies.

## Discussion

This study presented a significant positive impact of integrated palliative care on the quality of life of cancer patients compared to standard care (SMD = 3.25; 95% CI: 1.20–5.30; p < 0.001).

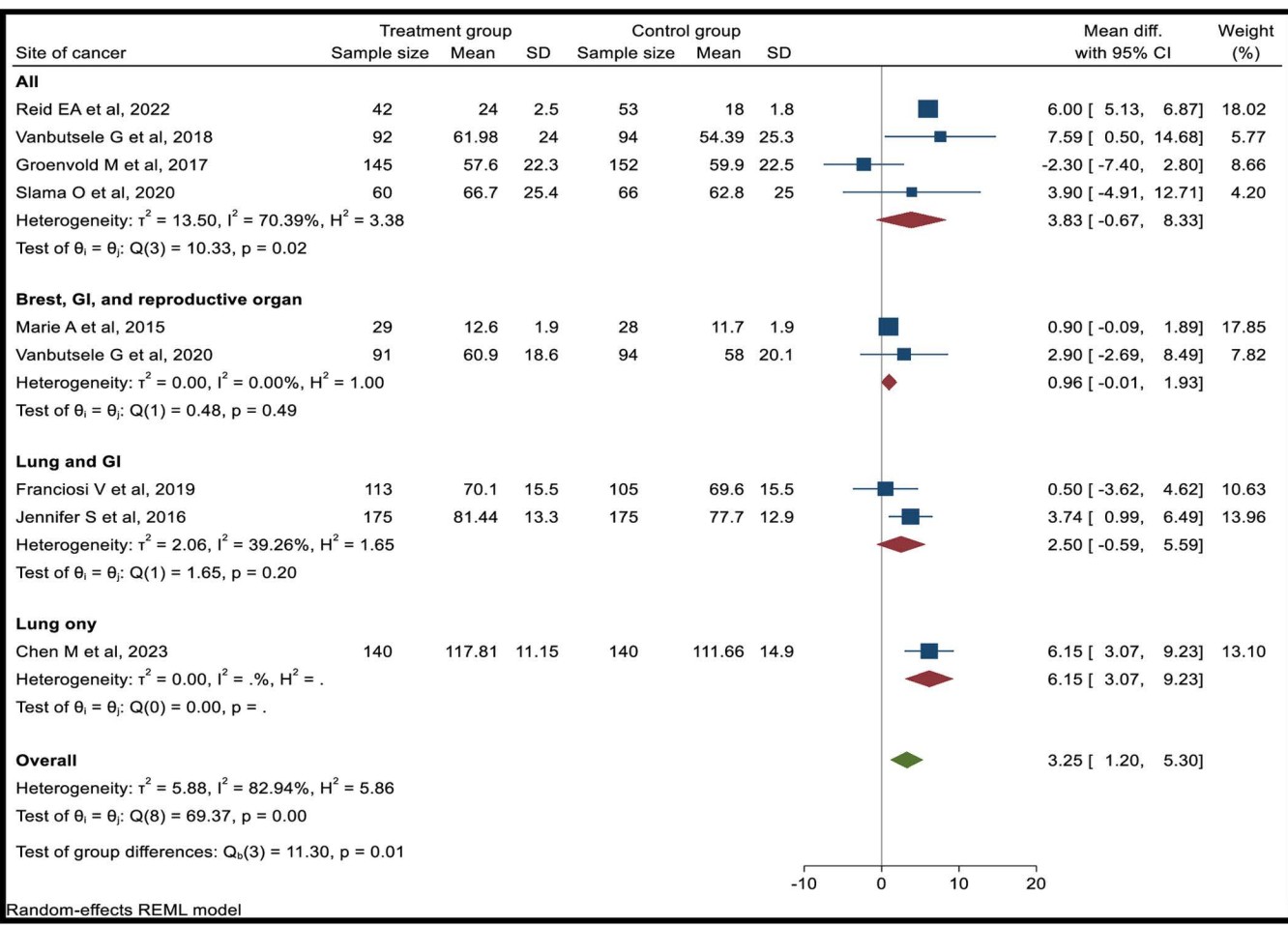

**Fig 4. Forest plot using a random-effects model illustrating sub-group analysis by site of cancer to show the comparative effects of integrated palliative care versus standard care on the quality of life in cancer patients.**

This is because IPC is attributed to its holistic approach, which addresses not only physical symptoms but also emotional, social, and spiritual needs. Integrated palliative care provides tailored support from a multidisciplinary team, offering pain and symptom management, psychological counseling, and assistance with treatment decisions [24]. This comprehensive care enhances patients' ability to cope with the physical and psychological burden of cancer, leading to improved overall well-being. Moreover, early intervention in symptom control and emotional support reduces distress and enhances quality of life, especially in high-burden cancers like lung cancer, where patients experience intense symptoms [2]. This approach contrasts with standard care, which primarily focuses on treating the disease and may overlook the broader quality-of-life issues faced by patients. This finding aligns with existing literature that consistently demonstrates the effectiveness of palliative care in improving various health-related quality of life (HRQOL) dimensions in cancer patients, including physical, emotional, and social well-being. Comparatively, a systematic review also reported significant HRQOL improvements with palliative care integration, but it highlighted that benefits were more consistent for psychosocial outcomes rather than physical functioning [25]. Similarly, Zimmermann *et al.* (2014) found that early integration of palliative care improved both

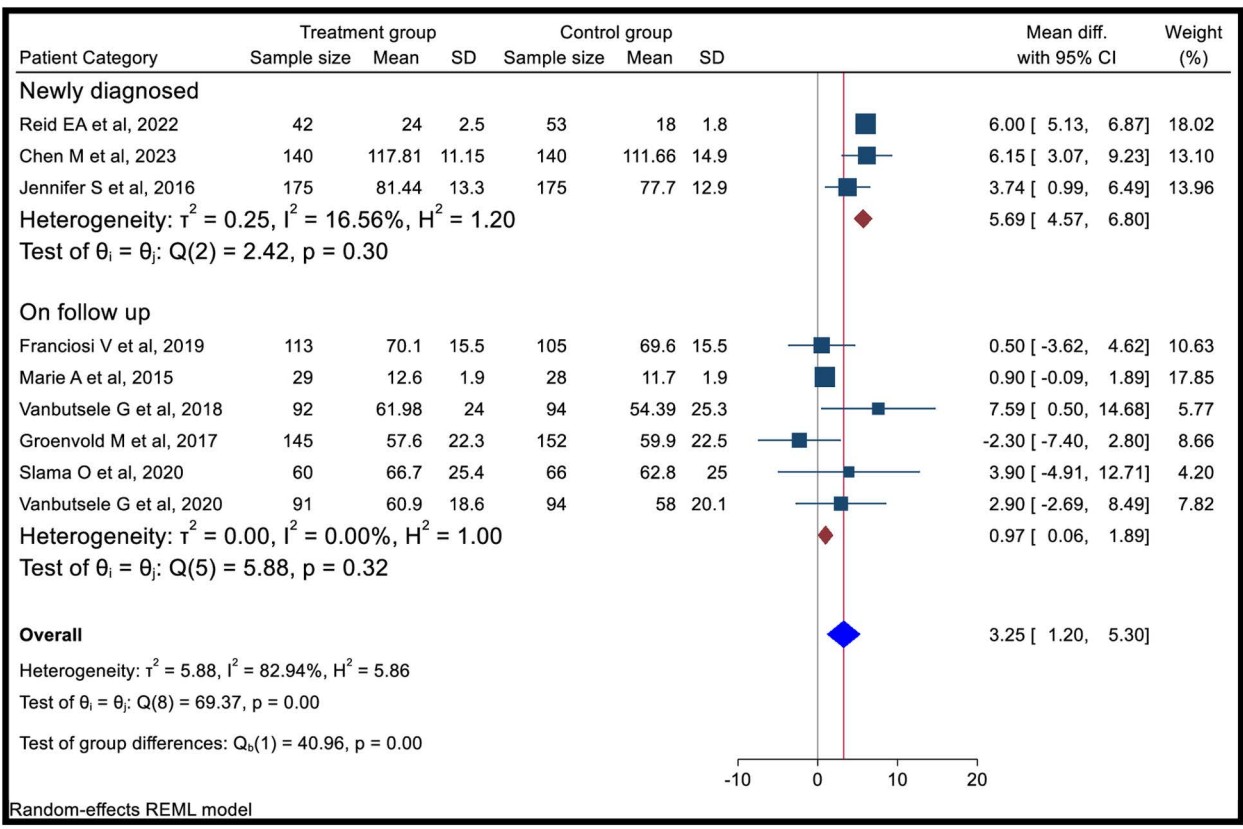

**Fig 5. Forest plot using a random-effects model illustrating sub-group analysis by patient category to show the comparative effects of integrated palliative care versus standard care on the quality of life in cancer patients.**

quality of life and mood, with lung cancer patients particularly benefiting from early interventions, corroborating the results of this study where lung cancer showed the highest SMD [26].

The overall assessment revealed that the risk of bias was predominantly low across the included RCTs. Most studies demonstrated rigorous randomization and allocation concealment processes, with effective blinding of participants, assessors, and personnel. These findings align with similar assessments conducted in prior systematic reviews, where comprehensive appraisal tools such as JBI and RoB 2 consistently contributed to improving the confidence in synthesized results [27]. The low risk of bias across studies suggests that the included trials provide a robust evidence base for assessing the comparative impact of integrated palliative care versus standard care on the quality of life in cancer patients.

Sub-group analysis showed that the impact of palliative care varied by region, with Asian and African studies reporting the highest standard mean difference. This could be attributed to the fact that, in many African and Asian countries, access to cancer care services, especially curative treatments such as radiotherapy and chemotherapy is often constrained by economic and infrastructural limitations [28]. Integrated palliative care programs help address this gap by prioritizing symptom control, psychological support, and end-of-life care, which significantly improve patients' quality of life [24]. Moreover, these programs in Africa and Asia often adopt a holistic approach, catering to the physical, emotional, social, and spiritual needs of patients. By incorporating culturally sensitive practices that align with local traditions and values, they enhance satisfaction for both patients and their families [29]. The type of cancer also impacted the outcomes, with lung cancer

| Follow up duration | Treatment group | | | Control group | | | | Mean diff. with 95% CI | Weight (%) |
|---|---|---|---|---|---|---|---|---|---|
| | Sample size | Mean | SD | Sample size | Mean | SD | | | |
| **Two months** | | | | | | | | | |
| Groenvold M et al, 2017 | 145 | 57.6 | 22.3 | 152 | 59.9 | 22.5 | | -2.30 [ -7.40, 2.80] | 8.66 |
| Heterogeneity: $\tau^2 = 0.00$, $I^2 = .\%$, $H^2 = .$ | | | | | | | | -2.30 [ -7.40, 2.80] | |
| Test of $\theta_i = \theta_j$: Q(0) = 0.00, p = . | | | | | | | | | |
| **Three months** | | | | | | | | | |
| Franciosi V et al, 2019 | 113 | 70.1 | 15.5 | 105 | 69.6 | 15.5 | | 0.50 [ -3.62, 4.62] | 10.63 |
| Marie A et al, 2015 | 29 | 12.6 | 1.9 | 28 | 11.7 | 1.9 | | 0.90 [ -0.09, 1.89] | 17.85 |
| Reid EA et al, 2022 | 42 | 24 | 2.5 | 53 | 18 | 1.8 | | 6.00 [ 5.13, 6.87] | 18.02 |
| Vanbutsele G et al, 2018 | 92 | 61.98 | 24 | 94 | 54.39 | 25.3 | | 7.59 [ 0.50, 14.68] | 5.77 |
| Jennifer S et al, 2016 | 175 | 81.44 | 13.3 | 175 | 77.7 | 12.9 | | 3.74 [ 0.99, 6.49] | 13.96 |
| Vanbutsele G et al, 2020 | 91 | 60.9 | 18.6 | 94 | 58 | 20.1 | | 2.90 [ -2.69, 8.49] | 7.82 |
| Heterogeneity: $\tau^2 = 5.12$, $I^2 = 85.57\%$, $H^2 = 6.93$ | | | | | | | | 3.35 [ 1.09, 5.61] | |
| Test of $\theta_i = \theta_j$: Q(5) = 61.63, p = 0.00 | | | | | | | | | |
| **Six months** | | | | | | | | | |
| Chen M et al, 2023 | 140 | 117.81 | 11.15 | 140 | 111.66 | 14.9 | | 6.15 [ 3.07, 9.23] | 13.10 |
| Slama O et al, 2020 | 60 | 66.7 | 25.4 | 66 | 62.8 | 25 | | 3.90 [ -4.91, 12.71] | 4.20 |
| Heterogeneity: $\tau^2 = 0.00$, $I^2 = 0.00\%$, $H^2 = 1.00$ | | | | | | | | 5.90 [ 2.99, 8.81] | |
| Test of $\theta_i = \theta_j$: Q(1) = 0.22, p = 0.64 | | | | | | | | | |
| **Overall** | | | | | | | | 3.25 [ 1.20, 5.30] | |
| Heterogeneity: $\tau^2 = 5.88$, $I^2 = 82.94\%$, $H^2 = 5.86$ | | | | | | | | | |
| Test of $\theta_i = \theta_j$: Q(8) = 69.37, p = 0.00 | | | | | | | | | |
| Test of group differences: $Q_b(2) = 7.60$, p = 0.02 | | | | | | | | | |

-10    0    10    20

Random-effects REML model

**Fig 6. Forest plot using a random-effects model illustrating sub-group analysis by follow up duration to show the comparative effects of integrated palliative care versus standard care on the quality of life in cancer patients.**

patients exhibiting the greatest improvement in quality of life. This could be because lung cancer is often diagnosed at advanced stages when curative treatment options are limited, and patients face a significant burden of severe symptoms [30]. Integrated palliative care effectively mitigates these symptoms through targeted approaches, such as pharmacological pain management, oxygen therapy for breathlessness, and psychological support, resulting in notable quality-of-life enhancements [25]. Moreover, lung cancer patients frequently endure a heightened psychological burden due to the stigma surrounding smoking and the disease's rapid progression. Integrated palliative care programs address these challenges by providing psychosocial support, counseling, and involving families, which helps reduce anxiety and depression while improving emotional well-being [31–33]. The results of this study also revealed that studies with a six-month follow-up period had the highest standardized mean difference. This could be because a longer follow-up period provides more time to observe the intervention's effects, leading to more robust and dependable outcome estimates. Moreover, extended follow-up periods can capture long-term effects that shorter durations may overlook, such as the sustainability of results or delayed negative outcomes [34].

Although subgroup, meta-regression, and sensitivity analyses were performed to address heterogeneity, the observed variation may be attributed to other factors such as healthcare

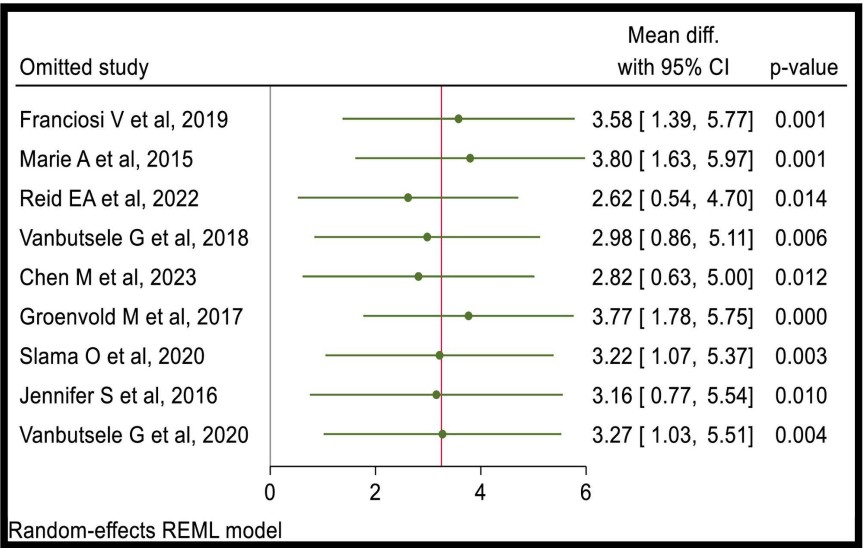

**Fig 7. Sensitivity analysis utilizing a random-effects model to demonstrate the impact of a leave-one-out estimate.**

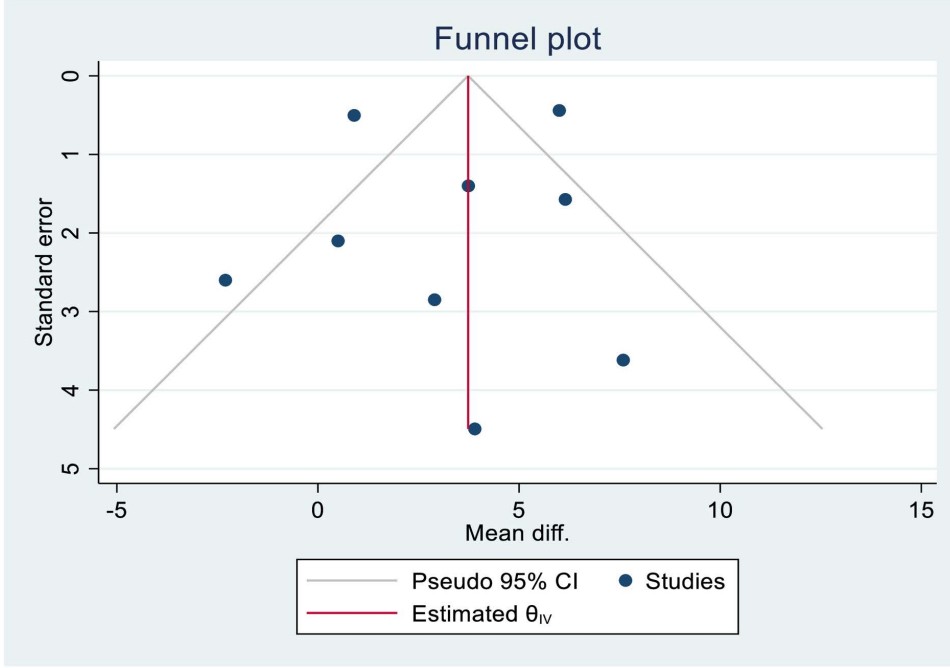

**Fig 8. Funnel plot with 95% confidence limits illustrating the distribution of the included RCTs in a study examining the comparative effects of integrated palliative care versus standard care on the quality of life in cancer patients.**

infrastructure, cultural attitudes, and demographic characteristics. In resource-limited settings, inadequate healthcare infrastructure often results in poor access to essential palliative care services such as pain management, psychosocial support, and trained personnel, thereby limiting improvements in QoL [28,35]. Cultural attitudes also play a significant role; in many Asian and African societies, discussing end-of-life care remains taboo, delaying palliative care interventions, while strong family involvement in these regions enhances holistic care that aligns with cultural values, improving satisfaction and outcomes [29,36]. Additionally, demographic factors such as age, socioeconomic status, and geographic disparities influence access to palliative care, with older, rural, and lower-income populations being disproportionately affected [25,37].

## Strengths and limitations of the study

A notable strength of this study is its comprehensive methodology, incorporating subgroup analysis, meta-regression, and sensitivity analysis to identify potential sources of heterogeneity. By accounting for moderators such as study setting, cancer type, and patient category, the study offers valuable insights into the differential impacts of integrated palliative care (IPC). The reliability of the findings is further reinforced by sensitivity analysis and the absence of significant publication bias. However, the study has certain limitations, including considerable heterogeneity ($I^2 = 82.94\%$) across the included studies, which may complicate the interpretation of the overall effect size. This heterogeneity could stem from differences in healthcare settings, such as resource availability, infrastructure, and access to IPC services, limiting the generalizability of the results. Furthermore, variability in IPC implementation, shaped by cultural, economic, and systemic factors, may also contribute to the observed outcome differences. Another limitation is the underrepresentation of studies from Asia and Africa, with only one study included from these regions. This limited sample size may have influenced the finding that IPC had a more significant impact in these regions, and caution is advised when interpreting this result given the potential for regional variability.

## Conclusion and recommendation

This study demonstrates a significant improvement in quality of life for cancer patients receiving integrated palliative care compared to standard care. Subgroup analysis revealed that the benefits are particularly notable in Asia and Africa, among lung cancer patients, and in newly diagnosed individuals, highlighting regional disparities in healthcare and the heavy symptom burden associated with certain cancer types. Prioritizing the early implementation of palliative care is essential to optimize patient outcomes and enhance quality of life across diverse populations. Policymakers can leverage these results to advocate for the integration of standardized palliative care protocols within existing cancer care systems, ensuring equitable access to IPC services, particularly in resource-limited settings. The study underscores the need to strengthen palliative care infrastructure to reduce disparities in care delivery and improve patient well-being. Future randomized controlled trials (RCTs) should focus on implementing standardized IPC protocols to reduce variability and explore which specific components, such as symptom control, psychosocial support, or spiritual care, yield the most significant improvements in quality of life.

## Supporting information

**S1 Text. S1 Fig.** The Cochrane RoB 2 tool provides a detailed assessment across various domains to compare the effects of integrated palliative care versus standard care on cancer patients' quality of life. **S1 Table.** The findings follow the Preferred Reporting Items for Systematic Review and Meta-analysis (PRISMA) guidelines, available as a supplementary file.

**S2 Table.** The Joanna Briggs Institute (JBI) Critical Appraisal Checklist is used to evaluate the comparative impact of integrated palliative care vs. standard care on the quality of life in cancer patients. **S3 Table.** Dataset for a study conducted on Comparative Impact of Integrated Palliative Care vs. Standard Care on the Quality of Life in Cancer Patients: A Global Systematic Review and Meta-Analysis of Randomized Controlled Trials. **S4 Table.** A numbered table of all studies identified in the literature search, including those that were excluded from the analyses. (ZIP)

## Author contributions

**Conceptualization:** Addisu Getie.

**Data curation:** Addisu Getie.

**Formal analysis:** Addisu Getie.

**Investigation:** Addisu Getie.

**Methodology:** Addisu Getie, Afework Edmealem.

**Software:** Addisu Getie, Tegene Atamenta Kitaw.

**Writing – original draft:** Addisu Getie.

**Writing – review & editing:** Addisu Getie, Afework Edmealem, Tegene Atamenta Kitaw.

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
