## [Decision Letter · Decision Letter 0]

11 Dec 2024

PONE-D-24-43910Comparative Impact of Integrated Palliative Care vs. Standard Care on the Quality of Life in Cancer Patients: A Global Systematic Review and Meta-Analysis of Randomized Controlled TrialsPLOS ONE

Dear Dr. Getie,

Thank you for submitting your manuscript to PLOS ONE. After careful consideration, we feel that it has merit but does not fully meet PLOS ONE’s publication criteria as it currently stands. Therefore, we invite you to submit a revised version of the manuscript that addresses the points raised during the review process.

We look forward to receiving your revised manuscript.

Kind regards,

Usama Waqar, M.B.B.S

Academic Editor

PLOS ONE

2. We note that your Data Availability Statement is currently as follows: [Not applicable]

3. As required by our policy on Data Availability, please ensure your manuscript or supplementary information includes the following:

Reviewers' comments:

Reviewer's Responses to Questions

**Comments to the Author**

1. Is the manuscript technically sound, and do the data support the conclusions?

Reviewer #1: Yes

Reviewer #2: Yes

Reviewer #3: Yes

2. Has the statistical analysis been performed appropriately and rigorously?

Reviewer #1: Yes

Reviewer #2: Yes

Reviewer #3: Yes

3. Have the authors made all data underlying the findings in their manuscript fully available?

Reviewer #1: Yes

Reviewer #2: Yes

Reviewer #3: No

4. Is the manuscript presented in an intelligible fashion and written in standard English?

Reviewer #1: Yes

Reviewer #2: Yes

Reviewer #3: Yes

5. Review Comments to the Author

Reviewer #1: The authors have conducted a systematic review and meta-analysis of randomized controlled trials (RCTs) assessing the Impact of integrated palliative care (IPC) in comparison to standard care (SC) on the quality of life (QoL) of different cancer patients across different stage and settings. The authors have reported that IPC is statistically associated with improved QoL, especially among lung cancer patients, patients from Africa, and newly diagnosed patients.

I have the following comments:

1. The authors emphasize the need for this study due to the lack of high-quality evidence supporting the integration of integrated palliative care (IPC) in cancer management, despite varying findings across studies. However, a literature search reveals a study by Xu S. et al., titled “The effects of integrated palliative care on quality of life and psychological distress in patients with advanced cancer: a systematic review and meta-analysis.” This study highlights the positive effects of IPC on patients' quality of life (QoL) and psychological well-being. It also examines multiple types of cancer, including hematological malignancies, similar to the author’s study. Additionally, it addresses variations in QoL improvement across different IPC models. The authors should acknowledge this evidence and clearly define the specific gap in the literature that their study aims to address.

2. The authors have mentioned in their “search strategy” that only articles published in English will be included. However, “eligibility criteria” mentions that there will be no restrictions on language. Similarly, authors mention in their “search strategy” that articles from inception will be included, while in “data collection and quality assessment” study mentions RCTs published in last ten years will be included. The authors need to address these inconsistencies.

3. The authors need to add definitions and explicit details to strengthen their study:

a. The authors need to clearly state the definition of IPC and SC which they used for their inclusion along with the IPC model they accepted.

b. The authors should also define the threshold they used to define high, moderate, or low risk of bias.

c. The authors need to define how they determined the advanced stage of cancer or if they accepted included study authors’ definition. Moreover, the authors have written “all” as the site of cancer in table 1 but did not explain clearly with a footnote that “all” defines as cancer involving lung, gastrointestinal, breast, and reproductive organs.

4. The authors need to add more details in table 1 to improve it including the phase of the RCT and the QoL tool(s) used by it.

5. While authors have explained their screening process along with two independent reviewer policy to reduce bias, the authors have failed to mention if a software, such as Covidence, was used for this process.

6. While the study methodology mentions the plan for risk of bias and quality assessment of included studies, the authors have not conducted and represented the quality assessment in results, explained them in discussion, or conducted associated analysis after excluding low quality RCTs.

7. Figure 3 represents USA’s standard mean difference (SMD) as significant with value of 4.84 [2.49-7.19]. However, authors have not represented this finding in their results or explained it in their discussion. Moreover, authors have not represented studies from Asia in their figure 3.

8. The authors have mentioned, “The studies consistently showed a positive impact of IPC on patient satisfaction, symptom management, and overall quality of life, especially in advanced cancer stages”. However, later it is mentioned, “These findings suggest that the integration of palliative care has a more pronounced impact on patients from Africa, those with lung cancer, and newly diagnosed individuals.” The authors should explain this finding in more detail.

9. The authors explain that African patients showed most benefit for IPC on QoL with other evidence also supporting more pronounced effects of palliative care in low-resource settings. However, this explanation is refuted by authors’ own positive finding for USA. The authors should highlight this and explain their findings.

10. The authors find that lung cancer is significantly associated with improved outcomes with evidence showing such trend among advanced lung cancer patients. However, the included RCT included newly diagnosed patients. The authors should highlight this difference. Moreover, they can cite a systematic review by Kochovska S et al. (PMID: 32953543) which also IPC use early in lung cancer.

Reviewer #2: Summary:

This manuscript provides a well-conducted systematic review and meta-analysis by analyzing RCTs across various geographic regions and cancer types. It provides a comprehensive and globally relevant perspective on IPC's benefits, specially in resource-limited settings. The study is methodologically rigorous and provides a valuable contribution to evidence-based practice in palliative oncology.

Revision:

1. While the authors conduct subgroup and sensitivity analyses to account for heterogeneity, a more detailed discussion on the variables leading to high heterogeneity (I² = 82.94%) could strengthen the interpretation, particularly around the variability in IPC implementation and patient demographics.

2. To improve clarity, explicitly define each subgroup for example patient categories (e.g., newly diagnosed vs. advanced) and summarize findings for each subgroup analysis.

3. Expand the limitations section to acknowledge that variations in healthcare settings and IPC implementation could limit generalizability of results

4. Proof read the manuscript for few minor grammatical errors and clarity

5. Suggest directions for future RCTs for example using a standardized IPC protocol to reduce variability and explore specific IPC components which has the greatest impact on QoL.

Reviewer #3: The authors conducted a systematic review and meta-analysis comparing the impact of integrated palliative care vs. standard care on the quality of life in cancer patients. The authors 9 randomized controlled trials showcasing an increase in quality of life with integrated palliative care. Stratified analysis by continent, patient population, and cancer type demonstrated consistency of the results. The authors concluded that through integrating palliative care, quality of life for all cancer patients can be improved significantly. I congratulate the authors on a well conducted study.

I have a few comments:

1. Authors should provide rationale for choosing only RCTs for their systematic review and meta-analysis.

2. Page 3, 3rd para of introduction, the authors mention that some studies show benefit of IPC while others do not. The reference of this statement is a single student showing benefit.

The authors have explained the knowledge gap and created the need for a systematic review. However, there is a need to cite studies showing no benefit of IPC considering the statement.

3. Several lines in the first para of discussion lack citations.

4. 2nd line of 2nd para of discussion citing a study by Davis et al. lacks a citation.

5. The authors should attempt to explain why low resource settings have a greater impact of integrated palliative care in the discussion section.

6. Similarly, in the para on page 10 explaining the impact of IPC on lung cancer should be expanded to talk briefly about the reasons behind this finding.

7. The authors should expand on the discussion section’s recommendation paragraph on future implications. How can the results of this study be used for policy making or improving cancer care guidelines? This has been briefly mentioned at the end of the manuscript but needs further elaboration.

8. The authors reported that the impact of IPC was more pronounced in Africa. There is only 1 study from Africa included in this review. Could this have influenced the results? Please include this in limitations section.

9. The authors should replace the word USA with North America if they are trying to compare continents (as mentioned in the manuscript several times)

10. The authors show a comparison of newly diagnosed vs. advanced cancer patients in figure 5. Would any of these newly diagnosed patients be considered advanced if they have Stage IV disease? This categorization needs rewording.

6. PLOS authors have the option to publish the peer review history of their article (what does this mean? ). If published, this will include your full peer review and any attached files.

**Do you want your identity to be public for this peer review?** For information about this choice, including consent withdrawal, please see our Privacy Policy .

Reviewer #1: No

Reviewer #2: No

Reviewer #3: No

---

## [Author Response · Author response to Decision Letter 1]

21 Dec 2024

Response to Reviewers

PONE-D-24-43910

Title: Comparative Impact of Integrated Palliative Care vs. Standard Care on the Quality of Life in Cancer Patients: A Global Systematic Review and Meta-Analysis of Randomized Controlled Trials

1. Thank you for submitting your manuscript to PLOS ONE. After careful consideration, we feel that it has merit but does not fully meet PLOS ONE’s publication criteria as it currently stands. Therefore, we invite you to submit a revised version of the manuscript that addresses the points raised during the review process.

Authors response 1: Thank you for your email regarding the review of our manuscript titled. We appreciate the time and effort of the reviewers and editorial team in evaluating our work and providing constructive feedback. We are pleased that you find our manuscript to have merit, and we are committed to addressing the reviewers' comments and concerns to enhance the quality and impact of our study. We carefully revised the manuscript and incorporated the feedback provided.

2. Please submit your revised manuscript by Jan 25 2025 11:59PM. If you will need more time than this to complete your revisions, please reply to this message or contact the journal office at plosone@plos.org. Authors response 2: Thank you for providing the timeline for submitting the revised version of our manuscript. We acknowledge the deadline of January 25, 2025, at 11:59 PM and we submitted the revised manuscript within this timeframe.

3.1. A rebuttal letter that responds to each point raised by the academic editor and reviewer(s). You should upload this letter as a separate file labeled 'Response to Reviewers'.

Authors response 3.1: Thank you for providing the detailed feedback from the academic editor and reviewers regarding our manuscript. We have carefully considered each comment and suggestion and have revised the manuscript accordingly to address the concerns raised and we attached our rebuttal Letter, titled “Response to Reviewers,” where we have provided a detailed, point-by-point response to all comments.

3.2. A marked-up copy of your manuscript that highlights changes made to the original version. You should upload this as a separate file labeled 'Revised Manuscript with Track Changes'.

Authors response 3.2: Thank you for the opportunity to revise our manuscript. We have made the necessary changes in response to the comments and suggestions provided by the academic editor and reviewers. As requested, we have prepared a marked-up version of the manuscript that highlights all revisions using track changes. This file is titled “Revised Manuscript with Track Changes” and has been uploaded as a separate document.

3.3. An unmarked version of your revised paper without tracked changes. You should upload this as a separate file labeled 'Manuscript'.

Authors response 3.3: Thank you for your instructions regarding the submission of our revised manuscript titled. As requested, we have prepared an unmarked version of the revised manuscript, with all track changes accepted, to reflect the final clean version. This file is titled “Manuscript” and has been uploaded as a separate document.

1. Please ensure that your manuscript meets PLOS ONE's style requirements, including those for file naming. The PLOS ONE style templates can be found at https://journals.plos.org/plosone/s/file?id=wjVg/PLOSOne_formatting_sample_main_body.pdf andhttps://journals.plos.org/plosone/s/file?id=ba62/PLOSOne_formatting_sample_title_authors_affiliations.pdf

Authors response 1: Thank you for the reminder regarding PLOS ONE’s style and file naming requirements. We ensured that our revised manuscript and all associated files comply fully with the journal's guidelines.

2. We note that your Data Availability Statement is currently as follows: [Not applicable]. Please confirm at this time whether or not your submission contains all raw data required to replicate the results of your study. Authors must share the “minimal data set” for their submission. PLOS defines the minimal data set to consist of the data required to replicate all study findings reported in the article, as well as related metadata and methods (https://journals.plos.org/plosone/s/data-availability#loc-minimal-data-set-definition). For example, authors should submit the following data:

The values behind the means, standard deviations and other measures reported;

The values used to build graphs;

The points extracted from images for analysis.

Authors do not need to submit their entire data set if only a portion of the data was used in the reported study. If your submission does not contain these data, please either upload them as Supporting Information files or deposit them to a stable, public repository and provide us with the relevant URLs, DOIs, or accession numbers. For a list of recommended repositories, please see https://journals.plos.org/plosone/s/recommended-repositories.

Authors response 2: Thank you for your reminder regarding the dataset. We would like to confirm that the authors have prepared a dataset in Microsoft Excel, which was specifically designed as an extraction sheet. This sheet was used to systematically collect and organize all relevant data from the studies included in our analysis. To ensure transparency and facilitate further understanding of our research process, we have provided this Microsoft Excel file as a relevant data repository document alongside our manuscript. Additionally, most of the data referenced in the manuscript can be found in Table 1. Please do not hesitate to reach out if additional clarifications or modifications to the dataset are required.

3. As required by our policy on Data Availability, please ensure your manuscript or supplementary information includes the following:

A numbered table of all studies identified in the literature search, including those that were excluded from the analyses.

For every excluded study, the table should list the reason(s) for exclusion.

If any of the included studies are unpublished, include a link (URL) to the primary source or detailed information about how the content can be accessed.

Authors response 3: Thank you for providing the instructions regarding the submission of a detailed table of studies identified in our literature search. In response, we have prepared a comprehensive document to meet your requirements. The document includes a numbered table summarizing all studies identified during the search, detailing both included and excluded studies and found in the relevant data repository document. However, due to the large volume of articles excluded for various reasons, it was not feasible to present all excluded studies in a single table. Instead, we have provided a representative summary, clearly listing the reasons for exclusion (e.g., not meeting inclusion criteria, irrelevant outcomes, duplicate data) to ensure transparency and adherence to systematic review reporting standards. Additionally, we confirm that our review did not include unpublished studies. Should further clarification or adjustments to the document be required, please feel free to let us know.

Name of data extractors and date of data extraction

Confirmation that the study was eligible to be included in the review.

All data extracted from each study for the reported systematic review and/or meta-analysis that would be needed to replicate your analyses.

Authors response 4: Thank you for your instructions regarding the submission of a detailed data extraction table. In response, we have prepared and uploaded a comprehensive table that includes all relevant information extracted from the primary research sources used in our systematic review and/or meta-analysis. The table addresses all specified requirements as follows, including the name of data extractor and date of extraction, study eligibility confirmation, and extracted data for analysis replication. The data extraction table has been uploaded as table file “Table 1”

5. If applicable for your analysis, a table showing the completed risk of bias and quality/certainty assessments for each study or outcome. Please ensure this is provided for each domain or parameter assessed. For example, if you used the Cochrane risk-of-bias tool for randomized trials, provide answers to each of the signaling questions for each study. If you used GRADE to assess certainty of evidence, provide judgements about each of the quality of evidence factor. This should be provided for each outcome.

Authors response 5: Thank you for your guidance regarding the submission of risk of bias and quality/certainty assessments for the studies included in our analysis. In response, we have diligently prepared the requested table, which includes the following: For the studies assessed using the Cochrane Risk-of-Bias tool (for randomized trials), we have provided responses to each signaling question for every study included in the review. The table clearly presents the risk of bias for each evaluated domain and is submitted as a supplementary file titled "Risk of Bias Assessment."

Authors response 6: Thank you for your inquiry regarding how we handled missing data in our analysis. We want to assure you that we carefully addressed the issue of missing data and implemented appropriate strategies to manage it, following established research practices. This included identifying and documenting missing data, as well as conducting sensitivity analyses to assess its impact.

Reviewers' comments:

Reviewer's Responses to Questions

Comments to the Author

1. Is the manuscript technically sound, and do the data support the conclusions?

Reviewer #1: Yes

Reviewer #2: Yes

Reviewer #3: Yes

Authors response 1: We appreciate your confirmation that our study adequately addresses the question, "Is the manuscript technically sound, and do the data support the conclusions?"

2. Has the statistical analysis been performed appropriately and rigorously?

Reviewer #1: Yes

Reviewer #2: Yes

Reviewer #3: Yes

Authors response 2: We appreciate your confirmation that our study adequately addresses the question, "has the statistical analysis been performed appropriately and rigorously?"

3. Have the authors made all data underlying the findings in their manuscript fully available?

Reviewer #1: Yes

Reviewer #2: Yes

Reviewer #3: No

Authors response 3: Thank you for approving the dataset. We confirm that the authors have prepared a Microsoft Excel file specifically designed as a data extraction sheet. This sheet was systematically used to collect and organize relevant data from the studies included in our analysis. To ensure transparency and enhance understanding of our research process, we have included this Excel file as a supplementary document alongside the manuscript. Additionally, most of the data referenced in the manuscript can be found in Table 1. Please feel free to contact us if further clarifications or modifications to the dataset are needed.

4. Is the manuscript presented in an intelligible fashion and written in standard English?

Reviewer #1: Yes

Reviewer #2: Yes

Reviewer #3: Yes

5. Review Comments to the Author

Reviewer #1: The authors have conducted a systematic review and meta-analysis of randomized controlled trials (RCTs) assessing the Impact of integrated palliative care (IPC) in comparison to standard care (SC) on the quality of life (QoL) of different cancer patients across different stage and settings. The authors have reported that IPC is statistically associated with improved QoL, especially among lung cancer patients, patients from Africa, and newly diagnosed patients.

I have the following comments:

1. The authors emphasize the need for this study due to the lack of high-quality evidence supporting the integration of integrated palliative care (IPC) in cancer management, despite varying findings across studies. However, a literature search reveals a study by Xu S. et al., titled “The effects of integrated palliative care on quality of life and psychological distress in patients with advanced cancer: a systematic review and meta-analysis.” This study highlights the positive effects of IPC on patients' quality of life (QoL) and psychological well-being. It also examines multiple types of cancer, including hematological malignancies, similar to the author’s study. Additionally, it addresses variations in QoL improvement across different IPC models. The authors should acknowledge this evidence and clearly define the specific gap in the literature that their study aims to address.

Authors response 1: Thank you for bringing the study by Xu S. et al. to our attention. We acknowledge that this work highlights the positive effects of Integrated Palliative Care (IPC) on quality of life (QoL) and psychological well-being in patients with advanced cancer and examines variations in QoL improvement across IPC models. However, while their study provides valuable insights, our review aims to address specific gaps that were not thoroughly explored, including the comparative impact of IPC versus standard care across diverse geographic settings and populations. To clarify the uniqueness of our study, we have now included a discussion of the findings by Xu S. et al. in the introduction and elaborated on how our review builds upon and complements their work. Specifically, our study focuses on synthesizing evidence from randomized controlled trials to evaluate the global impact of IPC on QoL, with a particular emphasis on regional variations and the inclusion of studies published after Xu S. et al.'s review. We have updated the manuscript accordingly and appreciate your suggestion to refine the gap statement, ensuring the d

---

## [Decision Letter · Decision Letter 1]

2 Mar 2025

PONE-D-24-43910R1Comparative Impact of Integrated Palliative Care vs. Standard Care on the Quality of Life in Cancer Patients: A Global Systematic Review and Meta-Analysis of Randomized Controlled Trials

PLOS ONE

Dear Dr. Getie,

Thank you for submitting your manuscript to PLOS ONE. After careful consideration, we feel that it has merit but does not fully meet PLOS ONE’s publication criteria as it currently stands. Therefore, we invite you to submit a revised version of the manuscript that addresses the points raised during the review process.

Please address the editorial comment on the follow-up intervals for QoL assessment in included studies.

We look forward to receiving your revised manuscript.

Kind regards,

Usama Waqar, M.B.B.S

Academic Editor

PLOS ONE

Journal Requirements:

Additional Editor Comments:

The mentioned outcome is QoL. Different studies may have measured QoL at different follow-up intervals. Did you focus on QoL at a specific follow-up interval after IPC vs. SC? How did you pool QoL data from multiple studies with different follow-up intervals in your meta-analyses? Please address in the Methods section in the Outcome heading or the statistical analysis plan.

Reviewers' comments:

Reviewer's Responses to Questions

**Comments to the Author**

1. If the authors have adequately addressed your comments raised in a previous round of review and you feel that this manuscript is now acceptable for publication, you may indicate that here to bypass the “Comments to the Author” section, enter your conflict of interest statement in the “Confidential to Editor” section, and submit your "Accept" recommendation.

Reviewer #2: All comments have been addressed

Reviewer #4: All comments have been addressed

2. Is the manuscript technically sound, and do the data support the conclusions?

Reviewer #2: Yes

Reviewer #4: Yes

3. Has the statistical analysis been performed appropriately and rigorously?

Reviewer #2: Yes

Reviewer #4: Yes

4. Have the authors made all data underlying the findings in their manuscript fully available?

Reviewer #2: Yes

Reviewer #4: Yes

5. Is the manuscript presented in an intelligible fashion and written in standard English?

Reviewer #2: Yes

Reviewer #4: Yes

6. Review Comments to the Author

Reviewer #2: (No Response)

Reviewer #4: I applaud the authors on their rigorous edits, which have satisfied any concerns that I may have had. I believe this manuscript makes a valuable contribution to the literature.

7. PLOS authors have the option to publish the peer review history of their article (what does this mean? ). If published, this will include your full peer review and any attached files.

**Do you want your identity to be public for this peer review?** For information about this choice, including consent withdrawal, please see our Privacy Policy .

Reviewer #2: No

Reviewer #4: No

---

## [Author Response · Author response to Decision Letter 2]

4 Mar 2025

Response to Reviewers

PONE-D-24-43910R1

Title: Comparative Impact of Integrated Palliative Care vs. Standard Care on the Quality of Life in Cancer Patients: A Global Systematic Review and Meta-Analysis of Randomized Controlled Trials

1. Thank you for submitting your manuscript to PLOS ONE. After careful consideration, we feel that it has merit but does not fully meet PLOS ONE’s publication criteria as it currently stands. Therefore, we invite you to submit a revised version of the manuscript that addresses the points raised during the review process.

Authors response 1: Thank you for your email regarding the review of our manuscript. We appreciate the time and effort of the reviewers and editorial team in evaluating our work and providing constructive feedback. We are pleased that you find our manuscript to have merit, and we are committed to addressing the reviewers' comments and concerns to enhance the quality and impact of our study. We carefully revised the manuscript and incorporated the feedback provided.

2. Please address the editorial comment on the follow-up intervals for QoL assessment in included studies.

Authors response 2: Thank you for your comment. We have addressed the follow-up intervals for QoL assessment in the included studies in the revised manuscript in the method section, providing clarification on the methods used to handle varying follow-up times.

3. Please submit your revised manuscript by Apr 16 2025 11:59PM. If you will need more time than this to complete your revisions, please reply to this message or contact the journal office at plosone@plos.org. Authors response 3: Thank you for providing the timeline for submitting the revised version of our manuscript. We acknowledge the deadline of Apr 16 2025 11:59PM and we submitted the revised manuscript within this timeframe.

4.1. A rebuttal letter that responds to each point raised by the academic editor and reviewer(s). You should upload this letter as a separate file labeled 'Response to Reviewers'.

Authors response 4.1: Thank you for providing the detailed feedback from the academic editor and reviewers regarding our manuscript. We have carefully considered each comment and suggestion and have revised the manuscript accordingly to address the concerns raised and we attached our rebuttal Letter, titled “Response to Reviewers,” where we have provided a detailed, point-by-point response to all comments.

4.2. A marked-up copy of your manuscript that highlights changes made to the original version. You should upload this as a separate file labeled 'Revised Manuscript with Track Changes'.

Authors response 4.2: Thank you for the opportunity to revise our manuscript. We have made the necessary changes in response to the comments and suggestions provided by the academic editor and reviewers. As requested, we have prepared a marked-up version of the manuscript that highlights all revisions using track changes. This file is titled “Revised Manuscript with Track Changes” and has been uploaded as a separate document.

4.3. An unmarked version of your revised paper without tracked changes. You should upload this as a separate file labeled 'Manuscript'.

Authors response 4.3: Thank you for your instructions regarding the submission of our revised manuscript titled. As requested, we have prepared an unmarked version of the revised manuscript, with all track changes accepted, to reflect the final clean version. This file is titled “Manuscript” and has been uploaded as a separate document.

Journal Requirements:

Authors response 1: Thank you for your feedback. We have thoroughly reviewed the reference list and made necessary corrections.

Additional Editor Comments:

2. The mentioned outcome is QoL. Different studies may have measured QoL at different follow-up intervals. Did you focus on QoL at a specific follow-up interval after IPC vs. SC? How did you pool QoL data from multiple studies with different follow-up intervals in your meta-analyses? Please address in the Methods section in the Outcome heading or the statistical analysis plan.

Authors response 2: Thank you, we have included the information in the Methods section under the Outcome heading as “To pool QoL data from studies with varying follow-up intervals, we used random-effects models, which effectively account for differences in follow-up durations and the inherent variability across studies. This statistical approach adjusts for the fact that each study may measure QoL at different time points, allowing for a more accurate estimation of the overall effect. By incorporating random effects, we were able to address both within-study and between-study variability, ensuring that our pooled estimate appropriately reflects the diversity of follow-up intervals and study designs”

Reviewers' comments:

Reviewer's Responses to Questions

Comments to the Author

1. If the authors have adequately addressed your comments raised in a previous round of review and you feel that this manuscript is now acceptable for publication, you may indicate that here to bypass the “Comments to the Author” section, enter your conflict-of-interest statement in the “Confidential to Editor” section, and submit your "Accept" recommendation.

Reviewer #2: All comments have been addressed

Reviewer #4: All comments have been addressed

Authors response 1: Thank you

2. Is the manuscript technically sound, and do the data support the conclusions?

Reviewer #2: Yes

Reviewer #4: Yes

Authors response 2: Thank you

3. Has the statistical analysis been performed appropriately and rigorously?

Reviewer #2: Yes

Reviewer #4: Yes

Authors response 3: Thank you

4. Have the authors made all data underlying the findings in their manuscript fully available?

Reviewer #2: Yes

Reviewer #4: Yes

Authors response 4: Thank you

5. Is the manuscript presented in an intelligible fashion and written in standard English?

Reviewer #2: Yes

Reviewer #4: Yes

Authors response 5: Thank you

6. Review Comments to the Author

Reviewer #2: (No Response)

Reviewer #4: I applaud the authors on their rigorous edits, which have satisfied any concerns that I may have had. I believe this manuscript makes a valuable contribution to the literature.

Authors response 6: Thank you

7. PLOS authors have the option to publish the peer review history of their article (what does this mean?). If published, this will include your full peer review and any attached files.

Do you want your identity to be public for this peer review? For information about this choice, including consent withdrawal, please see our Privacy Policy.

Reviewer #2: No

Reviewer #4: No

---

## [Editor Report · Decision Letter 2]

5 Mar 2025

PONE-D-24-43910R2Comparative Impact of Integrated Palliative Care vs. Standard Care on the Quality of Life in Cancer Patients: A Global Systematic Review and Meta-Analysis of Randomized Controlled TrialsPLOS ONE

Dear Dr. Getie,

Thank you for submitting your manuscript to PLOS ONE. After careful consideration, we feel that it has merit but does not fully meet PLOS ONE’s publication criteria as it currently stands. Therefore, we invite you to submit a revised version of the manuscript that addresses the points raised during the review process. Please address the editorial comment.

We look forward to receiving your revised manuscript.

Kind regards,

Usama Waqar, M.B.B.S

Academic Editor

PLOS ONE

Additional Editor Comments:

Random effects model only accounts for heterogeniety, which refers to differences in effects, i.e., odds ratios, risk ratios, etc, being estimated in the different studies included in a meta-analysis. It does not account for differences in follow-up intervals of individual studies.

Please see the following text from the Cochrane Handbook: 9.5.4 Incorporating heterogeneity into random-effects models.

"A random-effects meta-analysis model involves an assumption that the effects being estimated in the different studies are not identical, but follow some distribution. The model represents our lack of knowledge about why real, or apparent, intervention effects differ by considering the differences as if they were random. The centre of this distribution describes the average of the effects, while its width describes the degree of heterogeneity."

If the authors have not accounted for differences in follow up interval, this is a substantial error with the study design. Studies with short-term follow up may underestimate impact of different modalities on QoL; they cannot be merged with studies assessing QoL at larger intervals straight up. The authors should report follow up intervals at which QoL was assessed by individual studies in their Characteristics of Included Studies Table, then perform either of the following approach that aligns mostly with the variations in follow up intervals in the included studies (cannot directly assess as the authors have not provided this data:

1. Stratified analysis by follow-up duration

2. Meta-regression incorporating follow-up interval as an effect size

3. Sensitivity analyses excluding studies with very short or very long follow up intervals and pooling data from studies with similar follow-up intervals.

---

## [Author Response · Author response to Decision Letter 3]

6 Mar 2025

Response to Reviewers

PONE-D-24-43910R1

Title: Comparative Impact of Integrated Palliative Care vs. Standard Care on the Quality of Life in Cancer Patients: A Global Systematic Review and Meta-Analysis of Randomized Controlled Trials

1. Thank you for submitting your manuscript to PLOS ONE. After careful consideration, we feel that it has merit but does not fully meet PLOS ONE’s publication criteria as it currently stands. Therefore, we invite you to submit a revised version of the manuscript that addresses the points raised during the review process.

Authors response 1: Thank you for your email regarding the review of our manuscript. We appreciate the time and effort of the reviewers and editorial team in evaluating our work and providing constructive feedback. We are pleased that you find our manuscript to have merit, and we are committed to addressing the reviewers' comments and concerns to enhance the quality and impact of our study. We carefully revised the manuscript and incorporated the feedback provided.

2. Please submit your revised manuscript by Apr 19 2025 11:59PM. If you will need more time than this to complete your revisions, please reply to this message or contact the journal office at plosone@plos.org. Authors response 2: Thank you for providing the timeline for submitting the revised version of our manuscript. We acknowledge the deadline of Apr 16 2025 11:59PM and we submitted the revised manuscript within this timeframe.

3.1. A rebuttal letter that responds to each point raised by the academic editor and reviewer(s). You should upload this letter as a separate file labeled 'Response to Reviewers'.

Authors response 3.1: Thank you for providing the detailed feedback from the academic editor and reviewers regarding our manuscript. We have carefully considered each comment and suggestion and have revised the manuscript accordingly to address the concerns raised and we attached our rebuttal Letter, titled “Response to Reviewers,” where we have provided a detailed, point-by-point response to all comments.

3.2. A marked-up copy of your manuscript that highlights changes made to the original version. You should upload this as a separate file labeled 'Revised Manuscript with Track Changes'.

Authors response 3.2: Thank you for the opportunity to revise our manuscript. We have made the necessary changes in response to the comments and suggestions provided by the academic editor and reviewers. As requested, we have prepared a marked-up version of the manuscript that highlights all revisions using track changes. This file is titled “Revised Manuscript with Track Changes” and has been uploaded as a separate document.

3.3. An unmarked version of your revised paper without tracked changes. You should upload this as a separate file labeled 'Manuscript'.

Authors response 3.3: Thank you for your instructions regarding the submission of our revised manuscript titled. As requested, we have prepared an unmarked version of the revised manuscript, with all track changes accepted, to reflect the final clean version. This file is titled “Manuscript” and has been uploaded as a separate document.

Additional Editor Comments:

Random effects model only accounts for heterogeniety, which refers to differences in effects, i.e., odds ratios, risk ratios, etc, being estimated in the different studies included in a meta-analysis. It does not account for differences in follow-up intervals of individual studies.

Please see the following text from the Cochrane Handbook: 9.5.4 Incorporating heterogeneity into random-effects models.

"A random-effects meta-analysis model involves an assumption that the effects being estimated in the different studies are not identical, but follow some distribution. The model represents our lack of knowledge about why real, or apparent, intervention effects differ by considering the differences as if they were random. The centre of this distribution describes the average of the effects, while its width describes the degree of heterogeneity."

If the authors have not accounted for differences in follow up interval, this is a substantial error with the study design. Studies with short-term follow up may underestimate impact of different modalities on QoL; they cannot be merged with studies assessing QoL at larger intervals straight up. The authors should report follow up intervals at which QoL was assessed by individual studies in their Characteristics of Included Studies Table, then perform either of the following approach that aligns mostly with the variations in follow up intervals in the included studies (cannot directly assess as the authors have not provided this data:

1. Stratified analysis by follow-up duration

2. Meta-regression incorporating follow-up interval as an effect size

3. Sensitivity analyses excluding studies with very short or very long follow up intervals and pooling data from studies with similar follow-up intervals.

Authors response: Thank you for your suggestion. We have now included the follow-up intervals for QoL assessments in the Characteristics of Included Studies Table (Table 1). Among the included studies, six had a 3-month follow-up period, two had a 6-month period, and one had a 2-month follow-up. Based on this data, we conducted a stratified analysis, which is presented in Fig 6, with the findings discussed accordingly. Additionally, we performed a meta-regression that incorporated the follow-up interval as an effect size, which can be found in the results section. Furthermore, a sensitivity analysis was conducted to account for variations in follow-up durations across studies.

---

## [Editor Report · Decision Letter 3]

9 Mar 2025

Comparative Impact of Integrated Palliative Care vs. Standard Care on the Quality of Life in Cancer Patients: A Global Systematic Review and Meta-Analysis of Randomized Controlled Trials

PONE-D-24-43910R3

Dear Dr. Getie,

We’re pleased to inform you that your manuscript has been judged scientifically suitable for publication and will be formally accepted for publication once it meets all outstanding technical requirements.

Kind regards,

Usama Waqar, M.B.B.S

Academic Editor

PLOS ONE
---

## [Editor Report · Acceptance letter]

PONE-D-24-43910R3

PLOS ONE

Dear Dr. Getie,

I'm pleased to inform you that your manuscript has been deemed suitable for publication in PLOS ONE. Congratulations! Your manuscript is now being handed over to our production team.

Kind regards,

on behalf of

Dr. Usama Waqar

Academic Editor

PLOS ONE